# Population Structure and Morphological Pattern of the Black-Spotted Pond Frog (*Pelophylax nigromaculatus*) Inhabiting Watershed Areas of the Geum River in South Korea

Jun-Kyu Park [1], Ki Wha Chung [1], Ji Yoon Kim [2] and Yuno Do [1,*]

1   Department of Biological Science, Kongju National University, Gongju 32588, Republic of Korea
2   Department of Biological Science, Kunsan National University, Gunsan 54150, Republic of Korea
*   Correspondence: doy@kongju.ac.kr; Tel.: +82-41-850-8501

**Abstract:** Black-spotted pond frogs (*Pelophylax nigromaculatus*), widely distributed in East Asia, can be suitably used for the study of population genetic patterns and ecosystem monitoring. To systematically manage, conserve, and study this species, it is necessary to understand its habitat range. We analyzed the genetic and morphological range of black-spotted pond frog populations within a watershed of the Geum River, one of the main rivers in South Korea. We genotyped the frogs based on seven microsatellite loci and defined the skull shape based on landmark-based geometric morphometrics. One watershed area was divided into 14 sub-watershed areas, the smallest unit of the Geum River basin. The genetic structure of frogs among the 14 sub-watershed areas did not differ significantly, nor was correlated with geographic distance. Therefore, frogs within these watershed areas constitute a single population. Morphologically, they differed between some sub-watershed areas, but morphological distance did not correlate with genetic distance but rather with geographic distance. This morphological change differs depending on the environmental gradient rather than the genetic structure. As a single population, frogs in this watershed area need to be managed in an integrated way. We suggest that the identification of response and adaptation by population genetics must be compared across and beyond the watershed range.

**Keywords:** black-spotted pond frog; management and study; morphological difference; population genetic range; watershed area

## 1. Introduction

Black-spotted pond frogs (*Pelophylax nigromaculatus*) are a common species and one of the easiest frogs to study in East Asia (Korea, Japan, China, and Russia) [1]. This frog is considered a suitable model species for genetic and ecological studies because they have low mobility and high philopatry between habitats and breeding sites, easily breeding in the laboratory and easy to sample in the field, and have high responsiveness to environmental factors in various types of habitats [2–8]. Therefore, black-spotted pond frogs have been used to understand genetic and ecological phenomena caused by various stressors, habitat conditions, and temporal change [5,6,9–11]. However, systematic studies on conservation and management have been carried out for endangered species such as the Suweon treefrog (*Hyla suweonensis*), the gold-spotted pond frog (*Pelophylax chosenicus*), and the narrow-mouth frog (*Kaloula borealis*) in South Korea, but not for the black-spotted pond frog. Although the focus of conservation biology has been on endangered species [12–14], the management and conservation of common species should not be neglected. Common species are equally susceptible to population decline or local extinction [15–17]. They may also have important ecological and functional roles over a wider spatiotemporal range than endangered species that exist in limited habitats and occur less frequently [17–19]. Common species can be important indicators for conservation and management roles because they have a wide geographic distribution and can inhabit diverse environmental

conditions [20,21]. Black-spotted pond frogs also have ecological, functional, and indicator significance, and thus, for their conservation and management, it is necessary to confirm their population range and define their spatiotemporal habitat.

Whether ecologically or genetically, identifying the range of a population can delineate its management unit [22]. Additionally, it helps manage populations more systematically by determining important or threatened habitats and extracting information about factors that connect or disconnect habitats [23,24]. The population range can be genetically inferred by analyzing genetic structure and diversity and gene flow [25–28]. Although many methods exist for analyzing population genetic patterns, microsatellite methods have high reproducibility compared to other analyses and are suitable for identifying fine-scale population structures [29,30]. Additionally, it is possible to identify the ecological traits or adaptations in habitats according to the population range by checking phenotypic differences according to these genetic structures. For instance, the skull is associated with feeding biology, predator defense, locomotory performance, and microhabitat use [31–34]. It also changes early in life history due to various factors that may control the timing of metamorphosis [35–37].

We determined the population structure of black-spotted pond frogs in the watershed area of the main river basin in South Korea and identified morphological differences according to genetic structure. Freshwater wetland ecosystems can be managed based on the watershed area, but this management range needs to be different for each taxon or trait of an organism. For some species, large or small streams connect habitats, whereas for other species, streams disconnect habitats [38–41]. Additionally, populations may be divided or unified according to the geographical structure associated with rivers, the history of rivers and inhabiting species, and the composition of the biogeographic area [11,25,27,42,43]. The hydrologic unit map is considered the basic hydrological system related to water cycles. The geographic range of the catchment area is set by identifying the confluence from the main stream of the river. In South Korea, these ranges are used as standard boundaries for use of water resources among water-related institutions and are considered as units of administration, conservation, and management. The largest unit, the watershed area of main rivers, is set around the natural independent river formed along the mountain range, and this area is divided into several watershed area, the outlet points where the natural streams join. Finally, each watershed area can be divided into sub-watershed areas, the smallest watershed units. Previously, we reported the genetic diversity and population genetic structure of black-spotted pond frogs in four main rivers [43]. However, no study has systematically analyzed the population structure by subdividing this watershed. The systematic analysis of population structure allows us to determine what stream or watershed size can genetically and morphologically separate populations of this species. It also helps to identify whether the genetic flow also changes with stream flow from upstream to downstream within the watershed area. Therefore, we confirmed the population genetic structure of black-spotted pond frogs in sub-watershed areas and identified morphological variations of the skull.

## 2. Materials and Methods

### 2.1. Field Investigation

We selected 14 sub-watershed areas from the Geum River Basin, which is one of the largest rivers in South Korea (Figure 1). Each sub-watershed area consisted of five collection sites. Adult male frogs were collected during their breeding season from May to July 2020. Three frogs were collected from each site resulting in a total of two hundred and ten frogs. The collected frogs were euthanized by pithing and stored in 70% ethanol. Muscle samples from the femur were collected, and skeletal morphology was analyzed using X-ray. All procedures of animal collection, experimentation, euthanasia, storage, and disposal were performed in accordance with the regulations and with the approval of the Experimental Animal Ethics Committee of Kongju National University (KNU_2019-01).

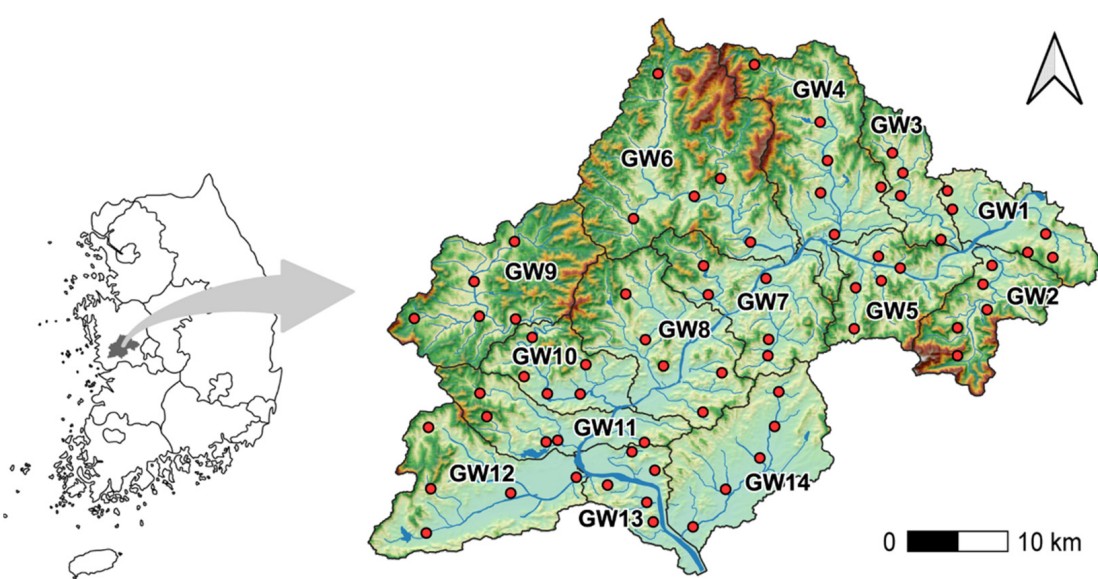

**Figure 1.** A total of 210 black-spotted pond frogs (*P. nigromaculatus*) were sampled from 70 sampling sites in 14 sub-watershed areas along the Geum river in South Korea. Black lines represent the boundaries of the sub-watershed areas. Blue lines indicate the tributaries and main rivers, and red points indicate the sampling sites.

## 2.2. DNA Extraction and Microsatellite Genotyping

Femur muscles were collected from 210 frogs to extract genomic DNA (gDNA). Total gDNA was extracted using DNeasy Blood and Tissue kits (Qiagen, Hilden, Germany) according to the manufacturer's instructions. The concentration and quality of gDNA in frogs were measured using a NanoDrop 2000 (Thermo Fisher Scientific, Wilmington, DE, USA). The measured gDNA was diluted to a concentration of 10–25 ng/μL. Seven polymorphic microsatellite loci were amplified using primers (Rnh-1, Rnh-2, Rnh-3, Rnh-4, Rnh-6, Rnh-10, and Rnh-12) and the protocol from a previous study [4]. We visualized the amplicons using a Seq-Studio Genetic Analyzer (Thermo Fisher-Applied Bio-systems, Foster City, CA, USA) and evaluated the dataset for genotype errors and the presence of null alleles using the GeneMapper version 6.1 (Thermo Fisher-Applied Bio-systems, Foster City, CA, USA).

## 2.3. Analysis of the Genetic Diversity and Population Structure

The genetic diversity and population structure of 210 *P. nigromaculatus* individuals were identified using the genotype dataset. The deviations of the Hardy-Weinberg equilibrium (HWE) and linkage disequilibrium (LD) of seven microsatellite loci were assessed using GENEPOP version 4.7 [44]. We did not detect null alleles, significant deviations from HWE, or evidence of LD at the seven loci. Additionally, seven loci were sufficient to establish population differentiation in 210 frogs (Figure 2).

The mean number of alleles ($N_A$), effective number of alleles ($N_E$), observed ($H_O$), and expected ($H_E$) heterozygosity, Shannon's information index (I), molecular diversity (h), and inbreeding coefficient relative to the subpopulation ($F_{IS}$) were calculated for all populations using GenAlEx version 6.5 add in Microsoft Excel [45] and Arlequin version 3.5 [46]. We selected I, h, and $F_{IS}$ to compare the genetic diversity of frogs in the 14 sub-watershed areas. Paired population differentiation (paired $F_{ST}$) and *p*-values obtained by Arlequin version 3.5 were used to compare the genetic distance among populations of 14 sub-watershed areas. Nei's genetic distance obtained by GenAlEx version 6.5 was used to create a hierarchical tree (unweighted pair group method with an arithmetic mean, UPGMA) in using the three software packages [47].

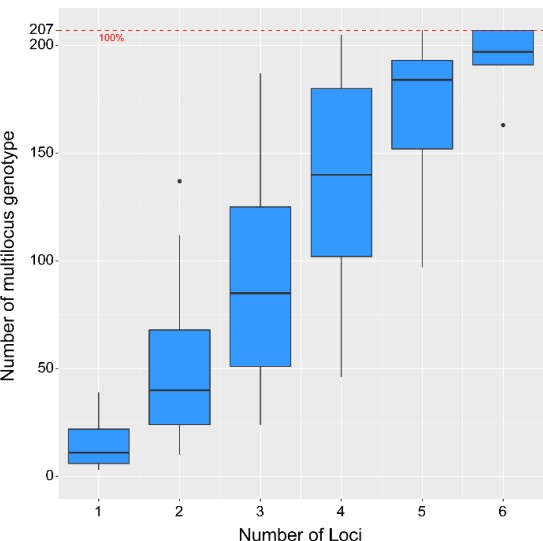

**Figure 2.** The number of multilocus genotypes (MLG) based on the number of combined loci from 210 black-spotted pond frogs. When we included six or more loci, the 207 individuals could be separated with a 100% accuracy. The central square dot indicates the mean value, the central line indicates the median value, the bottom box indicates the 25th percentile value, the top box indicates the 75th percentile value, the bottom and top of line indicate the range within 1.5 interquartile, and the circles indicate the outlier.

Bayesian clustering algorithms were used to identify the population genetic structure of frogs among 14 sub-watershed areas using STRUCTURE version 2.3.4 [48]. STRUCTURE analysis, an admixture model, confirms whether individual *i* has inherited a portion of its genetic material from ancestors in population *k*. One hundred thousand simulations were included in each analysis after an initial burn-in of 100,000 simulations. The STRUCTURE Harvester [49] estimated the best K value in the range of 1 to 14 possible clusters with three independent runs each. This was based on the second-order rate of change in the log probability of the data between successive values of K. We used the ΔK method [50] to identify the best-supported K value, which was determined based on the K value with the highest change ratio in the posterior probabilities of the two sequential K values. The STRUCTURE harvester showed that the most suitable K for dividing 14 populations was two (Figure 3).

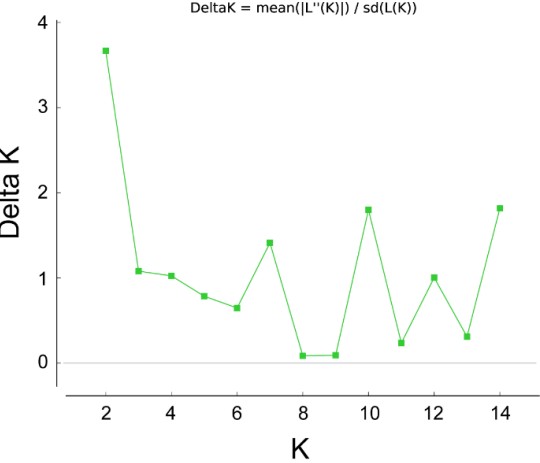

**Figure 3.** Best K value determination using the STRUCTURE harvester. One hundred thousand simulations after burn-in of 100,000 simulations were performed for the ΔK (delta K) method. Three independent runs in the range of 2 to 14 possible clusters were obtained by the simulations. The highest delta K value was achieved with a K of 2.

Discriminant analysis of principal components (DAPC) was used to compare the genetic structure of frogs. This multivariate clustering method used in the 'adegenet' R package [51]. Principal component analysis (PCA) was performed to reduce the dimension of genetic variation in DAPC. The linear combination of correlated alleles in the linear discriminant analysis was then produced using principal components.

### 2.4. Bone Image Tomography

To analyze skull morphology, we performed X-ray tomography as previously described [52,53]. We obtained skeletal images using fixed frog specimens in the same position. A Styrofoam plate was fixed at the bottom of a plastic box containing 70% ethanol. The frogs were placed on top and fixed with pins. The pins were not directly pierced through the frogs. After three days, the frogs were completely fixed, labeled, and stored in a separate bottle. A dual-energy X-ray absorptiometer (DEXA; Medikors InAlyzer, Seong-Nam, Republic of Korea), at the Korea Basic Science Institute (Gwangju, Korea), was used to obtain the skeleton images, which were then used to analyze the skull shapes.

### 2.5. Comparison of Skull Shapes

The skull shape was analyzed using landmark-based geometric morphometrics. We used the TpsDig software [54] to digitize and designate eight landmark points in the skull (Figure 4) according to a previous study performed on the same species [6,34]: (1) the posterior tip of the premaxilla, (2) the left tip of the maxilla, (3) the anterior tip of the sphenethmoid, (4) the right tip of the maxilla, (5) the left tip of the quadrate, (6) the right tip of the quadrate, (7) the left posterior tip of the prootic, and (8) the posterior right tip of the prootic. The digitized landmark coordinates were converted into Procrustes coordinates using the Morpho J software (version 1.07a, Manchester, UK). Canonical variate analysis (CVA) was used to compare differences in skull shape among frogs. The Procrustes distance and significance of differences in skull shape between frogs were obtained after 1000 permutation rounds in CVA. The contribution was calculated from each canonical variate (CV). The differences in skull shapes were visualized using a rectangular grid with landmark vectors and a wireframe graph of CV1 and CV2.

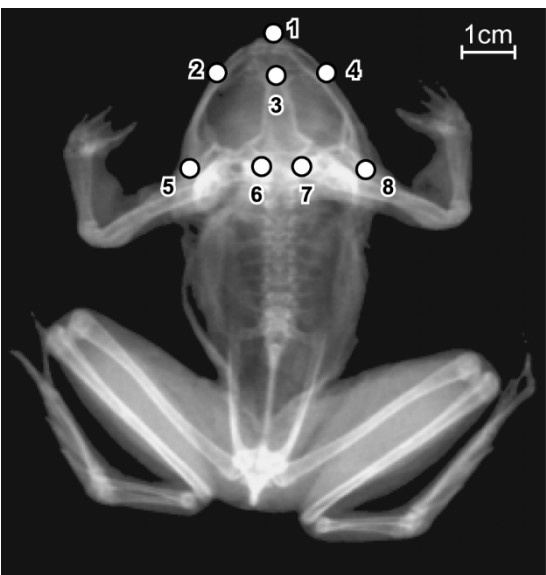

**Figure 4.** X-ray body image of a black-spotted pond frog obtained by a dual-energy X-ray absorptiometry (DEXA). Eight landmark points characterize the skull shape: (1) the posterior tip of the premaxilla, (2) the left tip of the maxilla, (3) the anterior tip of the sphenethmoid, (4) the right tip of the maxilla, (5) the left tip of the quadrate, (6) the right tip of the quadrate, (7) the posterior left tip of the prootic, and (8) the posterior right tip of the prootic.

*2.6. Correlation Analysis of Genetic, Morphological, and Geographic Distance among 70 Collecting Sites*

The paired $F_{ST}$ among 70 collection sites in 14 sub-watershed areas were obtained using Arlequin version 3.5. Procrustes distances among 70 collection sites were obtained using the Morpho J software (version 1.07a, Manchester, UK). Geographic distances among 70 collection sites were computed to find evidence of genetic and morphological separation by geographic isolation. The matrix of geographic distance was obtained by calculating the distance between the geographic coordinates of collection sites from the 'geodist' R package [55]. These three-matrix data (genetic, morphological, and geographic distance) were used to analyze the correlation using the Mantel test (number of permutations 999) with the 'vegan' R package [56].

## 3. Results

*3.1. Comparison of Genetic Diversity and Population Genetic Structure*

The overall Shannon's information index (I) of the 210 frogs was 0.428 (0.377–0.524), the overall molecular diversity (h) was 0.442 (0.390–0.541), and the overall inbreeding coefficient relative to the subpopulation ($F_{IS}$) was 0.170 (−0.029–0.415). The genetic diversity of the GW12 population was highest in I and h and lowest in the GW7 population (Table 1). Except for four populations (GW12, GW10, GW14, and GW5), the genetic diversity was lower in most populations compared to the overall population. Most populations had a positive $F_{IS}$ value, whereas only the GW4 population had a negative $F_{IS}$ value.

**Table 1.** Genetic diversity of black-spotted pond frogs (*P. nigromaculatus*) from 14 sub-watershed areas based on seven microsatellite loci: Shannon's information index (I), molecular diversity (h), inbreeding coefficient relative to the population.

| Basin | N | I | h | $F_{IS}$ |
|---|---|---|---|---|
| GW1 | 15 | 0.386 | 0.399 | 0.329 |
| GW2 | 15 | 0.392 | 0.405 | 0.079 |
| GW3 | 15 | 0.404 | 0.418 | 0.051 |
| GW4 | 15 | 0.408 | 0.422 | −0.029 |
| GW5 | 15 | 0.448 | 0.463 | 0.343 |
| GW6 | 15 | 0.414 | 0.428 | 0.119 |
| GW7 | 15 | 0.377 | 0.390 | 0.026 |
| GW8 | 15 | 0.440 | 0.455 | 0.415 |
| GW9 | 15 | 0.405 | 0.419 | 0.168 |
| GW10 | 15 | 0.444 | 0.459 | 0.057 |
| GW11 | 15 | 0.420 | 0.434 | 0.103 |
| GW12 | 15 | 0.524 | 0.541 | 0.272 |
| GW13 | 15 | 0.427 | 0.441 | 0.179 |
| GW14 | 15 | 0.500 | 0.517 | 0.271 |
| Mean of total | | 0.428 | 0.442 | 0.170 |

The hierarchical tree (UPGMA) of the Nei's genetic distance separated the population of the 14 sub-watershed areas into two major groups (Figure 5a), consistent with the results of the STRUCTURE harvester. However, the STRUCTURE analysis revealed that the genetic structure of frogs was similar among 14 sub-watershed areas (Figure 5b). The 210 frogs were grouped into the same K-cluster, indicating that genetically, all frogs belonged to the same population.

The DAPC and STRUCTURE analysis showed that all frogs belonged to the same genetic population. Discriminant function 1 (DF1) explained 21.78% and DF2 16.76% of the total genetic variation. On the DF1 axis, the GW12 and GW11 populations were separated from the other populations. In contrast, on the DF2 axis, not all populations were separated by genetic variation (Figure 6a). Finally, the fine-scale genetic structure was revealed using 100% of the explanatory power. None of the discriminant functions were detected in the genetic difference of frogs from 14 sub-watershed areas (Figure 6b).



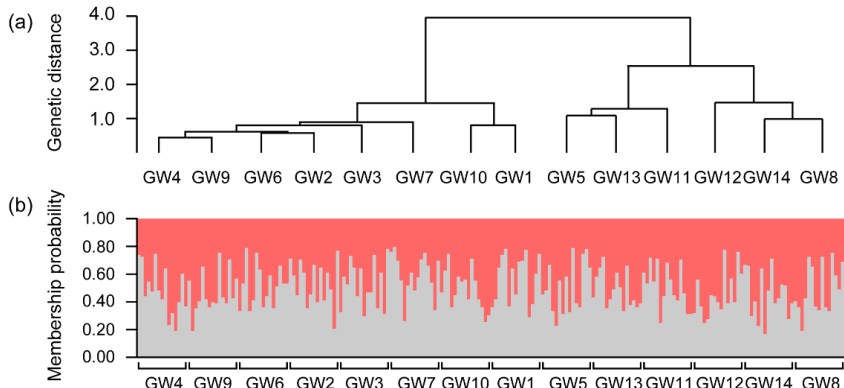

**Figure 5.** Genetic distance and population genetic structure of 210 black-spotted pond frogs from 14 sub-watershed areas: (**a**) unweighted pair group method with arithmetic mean (UPGMA) tree with Euclidean distance; (**b**) population genetic structure provided by STRUCTURE analysis.

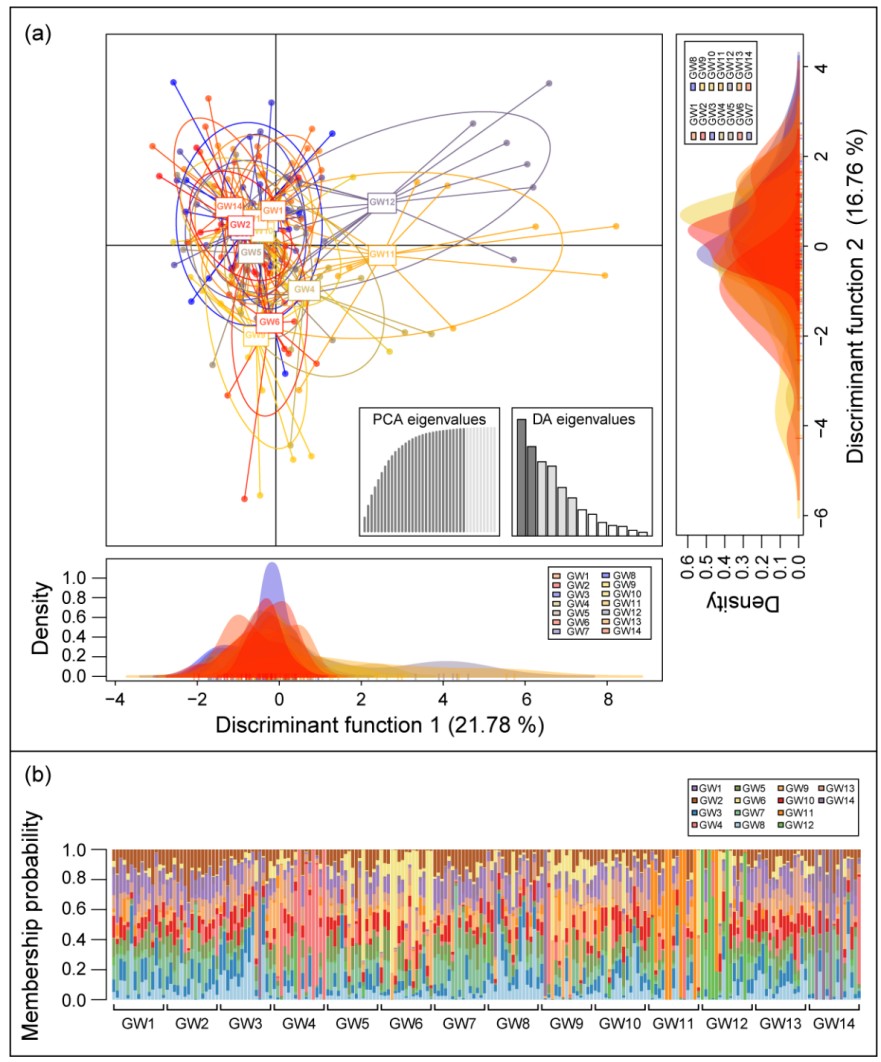

**Figure 6.** Population genetic structure based on discriminant analysis of principal components (DAPC): (**a**) scatter plot based on two major discriminant functions (DF). DF1 explained 21.78% of the genetic variation in black-spotted pond frogs, while DF2 explained 16.76%. Each node represents the genotype of frogs connected to a centroid assigned based on the clustering of the DAPC scores; (**b**) membership probability of DF determined that the sampled frogs were optimally clustered into 14 sub-watershed areas.

Based on paired $F_{ST}$, the GW11 population was genetically significantly distant from the eight populations (GW3, GW4, GW6, GW7, GW9, GW10, GW13, and GW14), indicating that it was the most genetically distant group (Figure 7). Additionally, GW4 and GW14 were genetically significantly distant from the four groups, although no specific pattern was found.

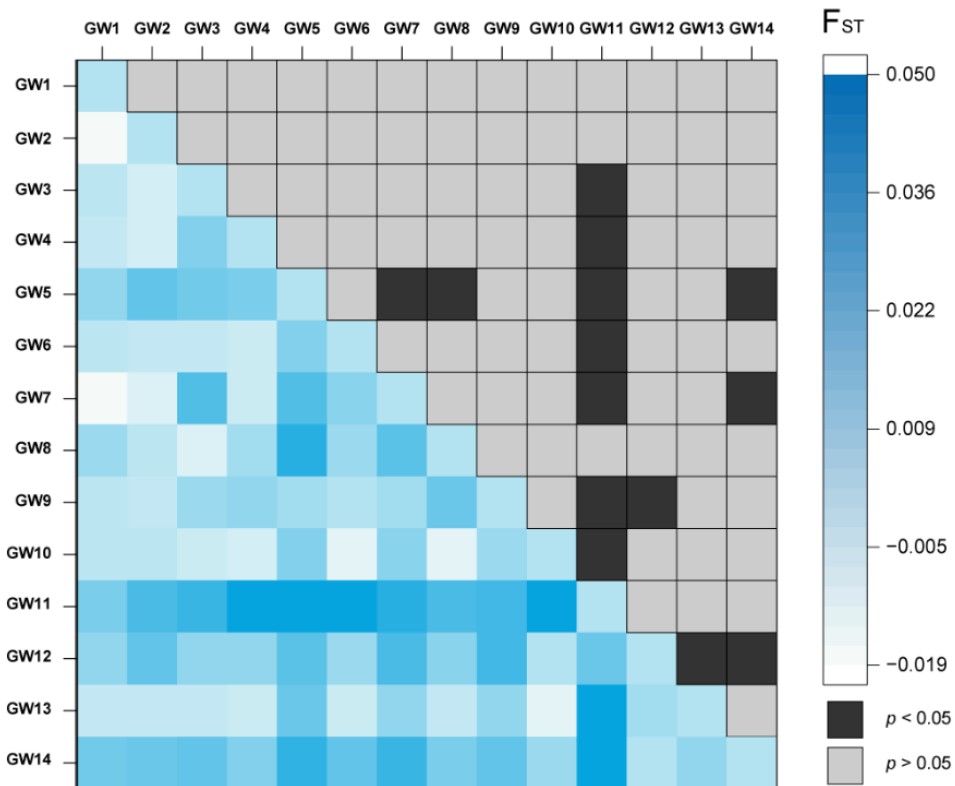

**Figure 7.** A heatmap showing paired $F_{ST}$ values among frogs from 14 sub-watershed areas. The blue boxes indicate paired $F_{ST}$ values. Dark gray colored boxes indicate the significant difference of $F_{ST}$ between two group, whereas light gray colored boxes indicate the non-significant difference of $F_{ST}$ between two groups.

### 3.2. Differences in the Skeletal Shape of the Skull

Two major morphological variations were identified using CVA. Canonical variate 1 (CV1) and CV2 explained 29.41% and 23.70% of the morphological variation, respectively, in 210 frogs from 14 populations. A higher CV1 value indicated a narrower and longer skull, while a higher CV2 values indicated a broader and shorter skull (Figure 8). Procrustes distance and significance of CVA distance showed that some populations were separated from the other groups (Figure 9). However, a relationship similar to the genetic segregation pattern was not observed.

### 3.3. Relationship between Geographic, Genetic, and Morphological Distances

Based on the geographical distances among the 70 collection sites, the Mantel test revealed no significant ($p > 0.05$) genetic isolation (Figure 10). Moreover, genetic distance did not correlate with morphological distance ($p > 0.05$), but morphological distance and geographic distance were strongly correlated (Mantel statistic, r = 0.160, $p < 0.05$).

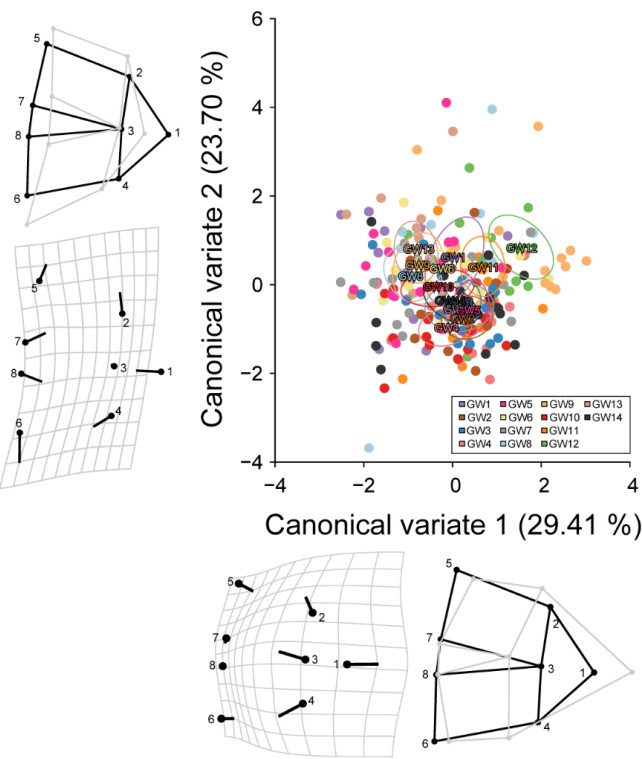

**Figure 8.** The skull shape variation among frogs collected from 14 sub-watershed areas determined by performing canonical variate analysis (CVA). The scatterplot consists of two major canonical variates (CV). CV1 and CV2 explained 29.41% and 23.70% morphological variation, respectively in frogs from 14 sub-watershed areas. Circles in the deformation grids and the black wireframe graphs indicate the skeletal shape of the individuals with the lowest CV values. Sticks in the deformation grids and the gray wireframe graphs indicate the change of skeletal shape with increasing CV value.

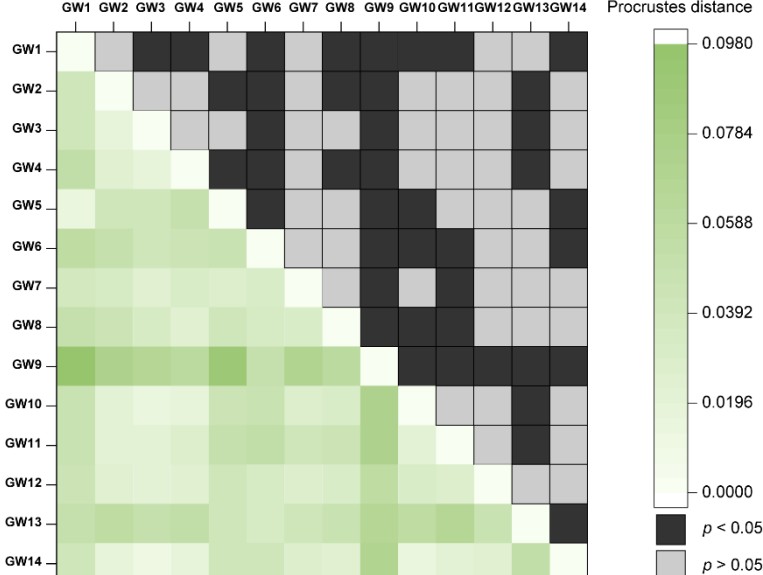

**Figure 9.** A heatmap showing Procrustes distance among frogs obtained from 14 sub-watershed areas. The green boxes indicate Procrustes distance values. Dark gray colored boxes indicate significant difference of Procrustes distance from CVA between two groups, whereas light gray colored boxes indicate the non-significant difference of Procrustes distance from CVA between two groups.

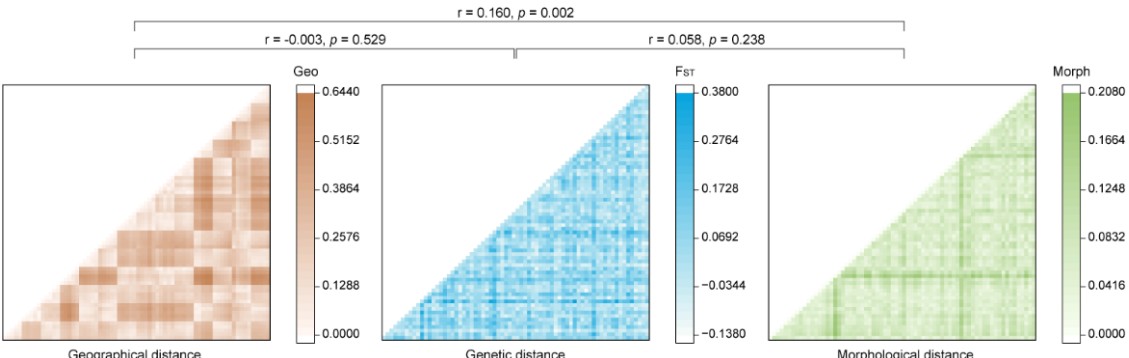

**Figure 10.** The relationship between the geographic, genetic, and morphological distance of 70 sampling sites in 14 sub-watershed areas. The geographic distance (Geo, in orange), paired population differentiation (paired $F_{ST}$, in blue), and the Procrustes distance (Morph, in green) are shown.

## 4. Discussion

South Korea's main rivers include the Geum, Han, Yeongsan, and Nakdong Rivers. These river basins are divided into several watersheds. It was confirmed that the black-spotted pond frogs were genetically distant along the main river basin [43]. We focused on the sub-watershed area that comprises the watershed area from one of the main rivers, the Geum River, and identified the population genetic structure in the smallest watershed unit. The results of Bayesian and multivariate clustering analyses showed that the black-spotted pond frogs inhabiting 14 sub-watershed areas in the Geum River genetically belong to the same population. However, populations inhabiting isolated sites or those with different habitat traits may be partially separated. Some populations showed high or low genetic diversity and differentiation of genetic structures in $F_{ST}$ and genetic diversity. Since we only wanted to determine whether the frogs were genetically divided within this watershed area, we do not know about differences depending on specific factors revealed from previous studies, such as isolation of habitats, biophysical connectivity, or landscape features [11,57,58]. The genetic segregation of black-spotted frogs in the four main river basins can be explained by geographic distances, large mountain ranges, and habitat differences [43,59]. In contrast, the Geum River watershed area targeted in our study is a single territory [42]. It is also geographically restricted, so the black-spotted pond frogs inhabiting this area belongs to a single population. Because of these results, it was not possible to predict or identify any impact on genetic flow by upstream and downstream, or on genetic isolation by the size or scale of the stream.

Contrary to the genetic patterns, we found morphological variations between frogs from the 14 sub-watershed areas. We expected morphological differences to change with genetic structure. However, these morphological distances correlated with geographic distance but not with genetic distance. Reportedly, the morphological change in head width or length can appear in response to the exposure duration of thyroid hormones due to accelerated or delayed metamorphosis [36,37]. The duration of frog metamorphosis can vary depending on environmental factors, such as duration of tadpole stage, predator pressure, temperature, protein content in food, and water level [36,60–64]. Even if we collected all frogs from paddy fields near streams with homogenous environments, it seems that the gradients of these environmental factors may differ among sub-watersheds. This difference in pattern may lead to an adaptive response to local selection pressures. Microsatellite patterns can be influenced by larger and/or stronger factors than detailed response, whereas morphological traits can be a sensitive response to local environmental changes [6,11,25]. The morphological response within the same population may be slightly different, and morphological comparisons of populations with different genetic structures must be performed in animals from a larger watershed range.

## 5. Conclusions

The population belonging to a single origin in a watershed area has several implications. From a species conservation and ecosystem management viewpoint, it is necessary to understand that the black-spotted pond frog population is a meta-population with a single origin and should be managed as a whole. In contrast, even in genetic structures with a single origin, a population with a remarkable difference in genetic structure or very low genetic diversity in some sites in a sub-watershed area to watershed area can be considered to experience isolation or differentiation or to be threatened by specific factors. In the future, it will be possible to identify populations exhibiting these patterns and manage them intensively. In particular, the black-spotted pond frog is considered to be in a condition that is experiencing population decline due to human agricultural pressures, habitat degradation, overharvesting, and chemical pollution, from the IUCN red list [65]. However, their conservation priority in Korea is very low. Therefore, it is necessary to organize such ecological information and manage the population based on it in preparation for decline. From an experimental and research viewpoint, the population within a watershed area can be regarded as a population with a single origin, and experiments can be conducted by excluding differences in population genetic structure. Conversely, to identify the differences in environmental response or adaptation according to the population's genetic structure, it is necessary to compare populations in a larger watershed area. Since some populations in our study showed segregated microscopic genetic structures, we expect to see more apparent separation when organisms from a larger area are included.

**Author Contributions:** Conceptualization, Y.D.; methodology, K.W.C.; software, J.-K.P.; validation, J.Y.K.; formal analysis, J.-K.P.; investigation, J.-K.P. and Y.D.; resources, K.W.C. and Y.D.; data curation, J.-K.P.; writing—original draft preparation, J.-K.P.; writing—review and editing, J.Y.K. and Y.D.; visualization, J.-K.P.; supervision, K.W.C.; project administration, J.Y.K. and Y.D.; funding acquisition, Y.D. All authors have read and agreed to the published version of the manuscript.

**Funding:** This research was supported by the National Research Foundation of Korea (NRF) grant funded by the Korean government (MSIT) (No. 2022R1A2C1004240), by Korea Environmental Industry & Technology Institute (KEITI) through Wetland Ecosystem Value Evaluation and Carbon Absorption Value Promotion Technology Development Project, funded by Korea Ministry of Environment (MOE) (2022003640001), and by the research grant of Kongju National University in 2022 (2022-0213-01).

**Institutional Review Board Statement:** Experimental procedures on animals were performed in accordance with regulations and with the approval of the Experimental Animal Ethics Committee of Kongju National University (KNU_2019-01).

**Informed Consent Statement:** Not applicable.

**Data Availability Statement:** Not applicable.

**Conflicts of Interest:** The authors declare no conflict of interest.

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
