# Peer review of "Population Structure and Morphological Pattern of the Black-Spotted Pond Frog (Pelophylax nigromaculatus) Inhabiting Watershed Areas of the Geum River in South Korea"

_sustainability, doi:10.3390/su142416530_

Round 1

Reviewer 1 Report

I read Park et al. MS, which provides a straightforward characterization of Pelophylax nigromaculatus subpopulations within its natural distribution in the Geum River drainage in South Korea. The authors used microsatellites for population structure analyses and skull geometric morphometrics for general anatomical comparisons. All 210 individuals from about 70 sampling sites seem to be part of the same population. As expected, the authors concluded that these individuals have little genetic and morphometric differentiation. Overall, this study is well done, yet there are not very exiting results to consider further explorations. I found the paragraph from lines 279-299 is too speculative given the results and the overall lack of evidence to consider environmental gradients. I will suggest the authors to remove such paragraph summarize and focus their discussion on the conservation threats for these frogs in the region of study: Are these frogs under human agricultural pressure in these study areas? What are the conservation priorities for these frogs now in South Korean and the Geum River in particular? Are these frogs being affect by climate change, ranaviruses, or Bd pandemic?

Here are some other suggestions:

Fig1: The blue dots that correspond to the localities sampled are not visible using the light blue hue. Please change to black or other contrasting color (e.g., red, yellow, etc).

Line 191: The sentence is awkward: ” …most populations compared to the than the overall population Most populations…”. Please rewrite.

Author Response

I read Park et al. MS, which provides a straightforward characterization of Pelophylax nigromaculatus subpopulations within its natural distribution in the Geum River drainage in South Korea. The authors used microsatellites for population structure analyses and skull geometric morphometrics for general anatomical comparisons. All 210 individuals from about 70 sampling sites seem to be part of the same population. As expected, the authors concluded that these individuals have little genetic and morphometric differentiation. Overall, this study is well done, yet there are not very exiting results to consider further explorations.

→ Thank you for the valuable comments of reviewer. We did our best to accept and correct the opinion of reviewer.

I found the paragraph from lines 279-299 is too speculative given the results and the overall lack of evidence to consider environmental gradients. I will suggest the authors to remove such paragraph summarize and focus their discussion on the conservation threats for these frogs in the region of study: Are these frogs under human agricultural pressure in these study areas? What are the conservation priorities for these frogs now in South Korean and the Geum River in particular? Are these frogs being affect by climate change, ranaviruses, or Bd pandemic?

→ We judged these sections to be too speculative. Therefore, we deleted these contents, made the context slicker, and reorganized the paragraph to focus on a different interpretation. These frogs are a dominant species of paddy fields and are closely related to human agriculture systems. Although there are few studies on climate change, ranaviruses, or Bd pandemics, they do not appear to be significantly affected by some existing studies. Instead, on the IUCN redlist, they are shown to be in decline and threatened by impacts such as agricultural pressures, habitat development, overharvesting, and pollution. Based on this content, we added a discussion focusing on conservation.

P11 L298-309: However, these morphological distances correlated with geographic distance but not with genetic distance. Reportedly, the morphological change in head width or length can be appeared in response to the exposure duration of thyroid hormones due to accelerated or delayed metamorphosis [36,37]. The duration of frog metamorphosis can vary depending on environmental factors, such as duration of tadpole stage, predator pressure, temperature, protein content in food, and water level [36,60-64]. Even if we collected all frogs from paddy fields near streams with homogenous environments, it seems that the gradients of these environmental factors may differ among sub-watersheds. This difference in pattern may lead to an adaptive response to local selection pressures. Microsatellite patterns can be influenced by larger and/or stronger factors than detailed response, whereas morphological traits can be sensitively response to local environmental changes [6, 11, 25]. The morphological response within the same population may be slightly different, and morphological comparisons of populations with different genetic structures must be performed in animals from a larger watershed range.

P12 L319-323: In particular, the black-spotted pond frog is considered to be a condition that is experiencing population decline due to human agricultural pressures, habitat degradation, overharvesting, and chemical pollution, from the IUCN red list [65]. However, their conservation priority in Korea is very low. Therefore, it is necessary to organize such ecological information and manage the population based on it in preparation for decline.

Here are some other suggestions:

Fig1: The blue dots that correspond to the localities sampled are not visible using the light blue hue. Please change to black or other contrasting color (e.g., red, yellow, etc).

→ We colored this point red.

Line 191: The sentence is awkward: ” …most populations compared to the than the overall population Most populations…”. Please rewrite.

→ We corrected this typo.

P6 L204-206: Except for four populations (GW12, GW10, GW14, and GW5), the genetic diversity was lower in most populations compared to the than the overall population. Most populations had a positive FIS value, whereas only the GW4 population had a negative FIS value.

Reviewer 2 Report

Population structure and morphological pattern of the black- 2 spotted pond frog (Pelophylax nigromaculatus) inhabiting watershed areas of the Geum River in South Korea

Overall:

I have to admit I am not a big fan of this kind of study as it seemed to be a ‘look and see’ project. When you are killing so many animals, I would be expecting a greater justification for why the study is necessary with more compelling apriori expectations. This is not the case. There is no conceptual framework to the study but just a statement that indicates the reason being to do the study is because it has not been done before. This is poor justification and one I cannot support. There is no doubt a lot of work has gone into this study and it is decently written and well-presented so on that side of things it is fine. But there are quite a few components where greater justification and more clarity on decision making process. Like the key one being how were the sub-watersheds selected and the randomisation of the sampling within those? And why would you expect any differences at all given the river system seems pretty well connected? If the goal was to understand if geographic distance is a factor then why sample all areas and not just areas that are closest and furthest apart? Or if there is an expected environmental gradient why did you not just sample the gradient. Before this study I would not have expected any differences, as it played out, but if you indicate some sites are geographically isolated then you may expect the difference. The work done seems fine based on my limited understanding of the DNA work but I strongly question the ethical value of this study.

Specific details:

Abstract:

Rewrite: “Black-spotted pond frogs (Pelophylax nigromaculatus), widely distributed in East Asia, are suitable in terms of population genetic patterns and ecosystem monitoring” – it makes no sense

Introduction:

Seems fair although I think the justification for why you would look at these frogs from a conservation perspective is relatively tenuous. It would have been enough to make a case for comparing the frogs across some environmental gradient because you had an apriori reason for some gradient affecting population structure and/or morphology. If not, then sacrificing so many animals seems unjustified to me. For that reason i would ask the authors to rewrite the conceptual framework in a matter that represents aprior expectations rather than just saying it should be done becuase it has not as yet.

I also found the conceptual framework lacking:

However, no study has systematically analyzed the population structure by subdividing this watershed.”  --- the answer to this is ‘so what?’ The work should have a compelling reason for why it is of interest and just because it has not been done is not enough.

Methods:

Can you give us some indication of how the sub-watersheds are decided upon? It is hard to know if these are just arbitrary? if they are arbitrary then why would you expect anything else but a random spread of characteristics? If they were selected based on pre-defined criteria then we need to know what they are and if that is the case then a model including those characteristics is apt.

I cannot talk to the molecular work in this study as it is not my expertise. BUT I would like some justification for the number of samples used and how that is expected to fulfil what the goal of the project was. So five collection sites of 3 frogs per watershed --- and is that enough power to ensure that the variation you are seeing at some sites is well represented and you are not just getting a difference in some places because some of the bigger ones just happened to be out that day? I am just not sure how many individuals you have to analyse to ensure what you are getting is truly a reflection of the variations at those points.

Geographic distances among 70 collection sites were computed to find evidence of genetic and morphological separation by geographic isolation. These three-matrix data (genetic, morphological, and geographic distance) were used to analyze the correlation using the Mantel test (number 182 of permutations 999) with the ‘vegan’ R package [55].” --- this seems very vague to me. How were geographic distances calculated and in what units? Or is this a relative scale like closer ones against more distance ones? It needs greater clarity.

Results:

Lots of cool figures

“Figure 7. A heatmap showing paired FST values among frogs from 14 sub-watershed areas. The numbers in the gray boxes indicate paired FST values between paired groups. Dark gray colored boxes indicate the significant paired FST and p-values, whereas light gray colored boxes indicate the non-significant paired FST and p-values.” – is this correct? Should it be the blue boxes indicate paired FST?

“Figure 9. A heatmap showing Procrustes distance among frogs obtained from 14 sub-watershed areas. The gray boxes indicate the Procrustes distances between paired groups. Dark gray colored boxes indicate significant Procrustes distance and p-values of CVA, whereas light gray colored boxes indicate the non-significant Procrustes distance and p-values of CVA.” – is this correct? Is it meant to be green boxes are Procrustes distances?

Discussion

The results of Bayesian and multivariate clustering analyses showed that the black-spotted pond frogs inhabiting 14 sub-watershed areas in the Geum River genetically belong to the same population. However, populations inhabiting isolated sites or those with different habitat traits may be partially separated.”

We expected morphological differences to change with genetic structure. Contrary to our expectations, the skull shapes correlated with the environmental gradients among the 14 sub-watersheds rather than genetic patterns.

The duration of frog metamorphosis can vary depending on environ-mental factors, such as duration of tadpole stage, predator pressure, temperature, protein content in food, and water level [36,59-63]. The gradients of these environmental factors differ between sub-watersheds.”

For these three phrases more information is needed – like what is mean by isolated sites here? Is that just geographic distance or true isolation i.e. no connectivity? And what habitat traits as this is said as if the different habitat traits are known and documented. Likewise in the second statement it suggests there is a correlation between environmental gradients and skull shapes?? Is there where is the proof of this as I do not see this in the text – maybe I missed it? Third statement categorically states the gradients of the environmental factors are different? Are they where is the information to back that up?

If there is no correlation between morphological and genetic distances is this an indication of potential plasticity? Or is it a consequence of sampling error in that the morphology of some groups is not the mean representative of those particular populations?

Author Response

Overall:

I have to admit I am not a big fan of this kind of study as it seemed to be a ‘look and see’ project. When you are killing so many animals, I would be expecting a greater justification for why the study is necessary with more compelling apriori expectations. This is not the case. There is no conceptual framework to the study but just a statement that indicates the reason being to do the study is because it has not been done before. This is poor justification and one I cannot support. There is no doubt a lot of work has gone into this study and it is decently written and well-presented so on that side of things it is fine. But there are quite a few components where greater justification and more clarity on decision making process. Like the key one being how were the sub-watersheds selected and the randomisation of the sampling within those? And why would you expect any differences at all given the river system seems pretty well connected?

→ The description for dividing watershed areas in South Korea have been added to introduction. Although river systems are well connected, they also act as geographic barriers to some organisms, as discussed in the introduction. For example, frogs can have a hard time crossing large main rivers. Mountain ranges also can be the same. We do not know how large the river must be in this species to act as a barrier or corridor. Additionally, the difference among watershed areas located relatively upstream and downstream from the sites of this study was also expected. However, no difference was found as a result. We have written this in more detail in the main text.

P2 L64-86: Freshwater wetland ecosystems can be managed based on the watershed area, but this management range needs to be different for each taxon or trait of an organism. For some species, large or small streams connect habitats, whereas, for other species, streams disconnect habitats [38-41]. Additionally, populations may be divided or unified according to the geographical structure associated with rivers, the history of rivers and inhabiting species, and the composition of the biogeographic area [11,25,27,42,43]. In South Korea, rivers are managed through a hydrologic unit map. The hydrologic unit map is consisted of considering the basic hydrological system related to water cycles. The geographic range of the catchment area is set by identifying the confluence from the main stream of the river. These range are used as standard boundaries for use of water resource among water related institutions and are considered as units of administration, conservation, and management. The largest unit, the watershed area of main rivers, is set around the natural independent river formed along the mountain range, and this area is divided into several watershed area, the outlet points where the natural streams join. Finally, each watershed area can be divided into sub-watershed areas, the smallest watershed units. Previously, we reported the genetic diversity and population genetic structure of black-spotted pond frogs in four main rivers [43]. However, no study has systematically analyzed the population structure by subdividing this watershed. The systematically analysis of population structure allows us to determine what stream or watershed size can genetically and morphologically separate populations of this species. It also helps to identify whether the genetic flow also changes with stream flow from upstream to downstream within the watershed area. Therefore, we confirmed the population genetic structure of black-spotted pond frogs in sub-watershed areas and identified morphological variations of the skull.

P11 L294-295: Because of these results, it was not possible to predict or identify any impact on genetic flow by upstream and downstream, or on genetic isolation by the size or scale of the stream.

If the goal was to understand if geographic distance is a factor then why sample all areas and not just areas that are closest and furthest apart?

→ Our main goal was not identification of genetic differentiation by geographic distance. The main purpose was to confirm whether there were genetic and morphological differences among watershed areas. Additionally, it was studied to confirm whether this was a geographical factor or a correlation with each other.

Or if there is an expected environmental gradient why did you not just sample the gradient.

→ → Likewise, the purpose of the study is to identify the range of genetic and morphological differences among watershed areas. However, we did not confirm in detail what caused this genetic integration and morphological difference. We thought this environment gradient was too speculative, so I modified it in the discussion section.

Before this study I would not have expected any differences, as it played out, but if you indicate some sites are geographically isolated then you may expect the difference.

→ We had no way to determine if some sites were isolated. So, we did this research. Several likely isolated sites have been identified, but further detailed studies are needed.

The work done seems fine based on my limited understanding of the DNA work but I strongly question the ethical value of this study.

→ This contents is included in “Institutional Review Board Statement” section. We added this contents in material and methods sections.

P2-3 L94-97: All procedures of animal collection, experimentation, euthanasia, storage and disposal was performed in accordance with the regulations and with the approval of the Experi-mental Animal Ethics Committee of Kongju National University (KNU_2019-01).

Specific details:

Abstract:

Rewrite: “Black-spotted pond frogs (Pelophylax nigromaculatus), widely distributed in East Asia, are suitable in terms of population genetic patterns and ecosystem monitoring” – it makes no sense

 → We modified this sentence.

P1 L9-10: Black-spotted pond frogs (Pelophylax nigromaculatus), widely distributed in East Asia, can be suitably used for study of population genetic patterns and ecosystem monitoring.

Introduction:

Seems fair although I think the justification for why you would look at these frogs from a conservation perspective is relatively tenuous. It would have been enough to make a case for comparing the frogs across some environmental gradient because you had an apriori reason for some gradient affecting population structure and/or morphology. If not, then sacrificing so many animals seems unjustified to me. For that reason i would ask the authors to rewrite the conceptual framework in a matter that represents aprior expectations rather than just saying it should be done becuase it has not as yet.

I also found the conceptual framework lacking:

However, no study has systematically analyzed the population structure by subdividing this watershed.”  --- the answer to this is ‘so what?’ The work should have a compelling reason for why it is of interest and just because it has not been done is not enough.

→ Although river systems are well connected, they also act as geographic barriers to some organisms, as discussed in the introduction. For example, frogs can have a hard time crossing large main rivers. Mountain ranges also can be the same. We do not know how large the river must be in this species to act as a barrier or corridor. Additionally, the difference among watershed areas located relatively upstream and downstream from the sites of this study was also expected. However, no difference was found as a result. We have written this in more detail in the main text.

P2 L64-86: Freshwater wetland ecosystems can be managed based on the watershed area, but this management range needs to be different for each taxon or trait of an organism. For some species, large or small streams connect habitats, whereas, for other species, streams disconnect habitats [38-41]. Additionally, populations may be divided or unified according to the geographical structure associated with rivers, the history of rivers and inhabiting species, and the composition of the biogeographic area [11,25,27,42,43]. In South Korea, rivers are managed through a hydrologic unit map. The hydrologic unit map is consisted of considering the basic hydrological system related to water cycles. The geographic range of the catchment area is set by identifying the confluence from the main stream of the river. These range are used as standard boundaries for use of water resource among water related institutions and are considered as units of administration, conservation, and management. The largest unit, the watershed area of main rivers, is set around the natural independent river formed along the mountain range, and this area is divided into several watershed area, the outlet points where the natural streams join. Finally, each watershed area can be divided into sub-watershed areas, the smallest watershed units. Previously, we reported the genetic diversity and population genetic structure of black-spotted pond frogs in four main rivers [43]. However, no study has systematically analyzed the population structure by subdividing this watershed. The systematically analysis of population structure allows us to determine what stream or watershed size can genetically and morphologically separate populations of this species. It also helps to identify whether the genetic flow also changes with stream flow from upstream to downstream within the watershed area. Therefore, we confirmed the population genetic structure of black-spotted pond frogs in sub-watershed areas and identified morphological variations of the skull.

Methods:

Can you give us some indication of how the sub-watersheds are decided upon? It is hard to know if these are just arbitrary? if they are arbitrary then why would you expect anything else but a random spread of characteristics? If they were selected based on pre-defined criteria then we need to know what they are and if that is the case then a model including those characteristics is apt.

→ The description for dividing watershed areas in South Korea have been added to introduction.

P2 L64-86: Freshwater wetland ecosystems can be managed based on the watershed area, but this management range needs to be different for each taxon or trait of an organism. For some species, large or small streams connect habitats, whereas, for other species, streams disconnect habitats [38-41]. Additionally, populations may be divided or unified according to the geographical structure associated with rivers, the history of rivers and inhabiting species, and the composition of the biogeographic area [11,25,27,42,43]. In South Korea, rivers are managed through a hydrologic unit map. The hydrologic unit map is consisted of considering the basic hydrological system related to water cycles. The geographic range of the catchment area is set by identifying the confluence from the main stream of the river. These range are used as standard boundaries for use of water resource among water related institutions and are considered as units of administration, conservation, and management. The largest unit, the watershed area of main rivers, is set around the natural independent river formed along the mountain range, and this area is divided into several watershed area, the outlet points where the natural streams join. Finally, each watershed area can be divided into sub-watershed areas, the smallest watershed units. Previously, we reported the genetic diversity and population genetic structure of black-spotted pond frogs in four main rivers [43]. However, no study has systematically analyzed the population structure by subdividing this watershed. The systematically analysis of population structure allows us to determine what stream or watershed size can genetically and morphologically separate populations of this species. It also helps to identify whether the genetic flow also changes with stream flow from upstream to downstream within the watershed area. Therefore, we confirmed the population genetic structure of black-spotted pond frogs in sub-watershed areas and identified morphological variations of the skull.

I cannot talk to the molecular work in this study as it is not my expertise. BUT I would like some justification for the number of samples used and how that is expected to fulfil what the goal of the project was. So five collection sites of 3 frogs per watershed --- and is that enough power to ensure that the variation you are seeing at some sites is well represented and you are not just getting a difference in some places because some of the bigger ones just happened to be out that day? I am just not sure how many individuals you have to analyse to ensure what you are getting is truly a reflection of the variations at those points.

→ In microsatellite study, the analysis that needs to pay attention to the number of samples is genetic diversity analysis. In the case of that sample size is about 5-10 individuals, a bias in the genetic diversity values may be seen in specific groups. In general, more than 20 individuals are considered stable in genetic diversity analysis (Refer). Our sample count is less than 20, but 15 seems stable enough. The appropriate sample size may also differ depending on the taxa or number of genetic markers. In insects, mammals, birds, and even within birds, optimal sample sizes vary among species (Refer). In the case of frogs, mobility is low, which may be reasonable, especially since this species is not a species that alternates between breeding and habitat. We tried to collect at the same time as much as possible, and all objects were collected within 2 months. However, it doesn't matter. Because genetic and morphological items are not response variables that can change rapidly in a short period of time like immune or physiological variables.

Geographic distances among 70 collection sites were computed to find evidence of genetic and morphological separation by geographic isolation. These three-matrix data (genetic, morphological, and geographic distance) were used to analyze the correlation using the Mantel test (number 182 of permutations 999) with the ‘vegan’ R package [55].” --- this seems very vague to me. How were geographic distances calculated and in what units? Or is this a relative scale like closer ones against more distance ones? It needs greater clarity.

→ We have added the more detailed information in the main text.

P6 L193-195: The matrix of geographic distance was obtained by calculating the distance between the geographic coordinates of collection site from the ‘geodist’ R package [55].

Results:

Lots of cool figures

“Figure 7. A heatmap showing paired FST values among frogs from 14 sub-watershed areas. The numbers in the gray boxes indicate paired FST values between paired groups. Dark gray colored boxes indicate the significant paired FST and p-values, whereas light gray colored boxes indicate the non-significant paired FST and p-values.” – is this correct? Should it be the blue boxes indicate paired FST?

→ We fixed this error.

Figure 7. A heatmap showing paired FST values among frogs from 14 sub-watershed areas. The blue boxes indicate paired FST values. Dark gray colored boxes indicate the significant difference of FST be-tween two group, whereas light gray colored boxes indicate the non-significant difference of FST between two group.

“Figure 9. A heatmap showing Procrustes distance among frogs obtained from 14 sub-watershed areas. The gray boxes indicate the Procrustes distances between paired groups. Dark gray colored boxes indicate significant Procrustes distance and p-values of CVA, whereas light gray colored boxes indicate the non-significant Procrustes distance and p-values of CVA.” – is this correct? Is it meant to be green boxes are Procrustes distances?

 → We fixed this error.

Figure 9. A heatmap showing Procrustes distance among frogs obtained from 14 sub-watershed areas. The green boxes indicate Procrustes distance values. Dark gray colored boxes indicate significant difference of Procrustes distance from CVA between two groups, whereas light gray colored boxes indicate the non-significant difference of Procrustes distance from CVA between two groups.

Discussion

The results of Bayesian and multivariate clustering analyses showed that the black-spotted pond frogs inhabiting 14 sub-watershed areas in the Geum River genetically belong to the same population. However, populations inhabiting isolated sites or those with different habitat traits may be partially separated.”

We expected morphological differences to change with genetic structure. Contrary to our expectations, the skull shapes correlated with the environmental gradients among the 14 sub-watersheds rather than genetic patterns.”

The duration of frog metamorphosis can vary depending on environ-mental factors, such as duration of tadpole stage, predator pressure, temperature, protein content in food, and water level [36,59-63]. The gradients of these environmental factors differ between sub-watersheds.”

For these three phrases more information is needed – like what is mean by isolated sites here? Is that just geographic distance or true isolation i.e. no connectivity? And what habitat traits as this is said as if the different habitat traits are known and documented.

→ The isolated population is a population with low genetic diversity and high genetic differentiation by geographic isolation or low habitat connectivity. It has nothing to do with geographic distance. No population was identified as an isolated population in our study.

Likewise in the second statement it suggests there is a correlation between environmental gradients and skull shapes?? Is there where is the proof of this as I do not see this in the text – maybe I missed it?

Third statement categorically states the gradients of the environmental factors are different? Are they where is the information to back that up?

→ We judged these sentences to be too speculative. Therefore, we deleted these contents, made the context slicker, and reorganized the paragraph to focus on a different interpretation.

If there is no correlation between morphological and genetic distances is this an indication of potential plasticity? Or is it a consequence of sampling error in that the morphology of some groups is not the mean representative of those particular populations?

→ It seems far from plasticity or sampling error. We did not assume that genes change morphology, but that if the genetic structure is different in the population, the skull morphology will also be different. This is a completely different mean. Microsatellite loci, which are generally short repetitive sequences, are often present in intronic regions rather than exon regions (genes) due to their high variability. Therefore, it is difficult to see that it has any genetic regulation of expression or genetic function. Of course, there are exceptions. Some studies or species also develop microsatellite markers in exon regions. The morphological plasticity we analyzed also responds only to environmental factors, independent of genetic differences. However, the multiple spatiotemporal factors within this watershed appear to be insufficient to completely independently modify genetic structure or morphological traits. This seems to have resulted in our results.

Reviewer 3 Report

The study is very interesting, as it is focused on the conservation and management of an endangered species. However, there are several comments:

1. The authors do not explain the way these animals were handled. Nothing about the guidelines followed for their protection. They neither mention if all procedures were approved by any ethic committee.

2. There are results about "Skeletal shape and lower body".  On the contrary, In Material and Methods it is clear that the authors obtain the skeleton images, which were used to analyze the skull shapes. Nothing about other different parts of the skeleton, that could be used for comparison or as landmarks.

3. The criteria to consider an animal as young or adult should be pointed out. 

4. English should be improved

Author Response

The study is very interesting, as it is focused on the conservation and management of an endangered species. However, there are several comments:

  1. The authors do not explain the way these animals were handled. Nothing about the guidelines followed for their protection. They neither mention if all procedures were approved by any ethic committee.

→ This contents is included in “Institutional Review Board Statement” section. We added this contents in material and methods sections.

P2-3 L94-97: All procedures of animal collection, experimentation, euthanasia, storage and disposal was performed in accordance with the regulations and with the approval of the Experi-mental Animal Ethics Committee of Kongju National University (KNU_2019-01).

  1. There are results about "Skeletal shape and lower body".  On the contrary, In Material and Methods it is clear that the authors obtain the skeleton images, which were used to analyze the skull shapes. Nothing about other different parts of the skeleton, that could be used for comparison or as landmarks.

→ We removed this typo.

  1. The criteria to consider an animal as young or adult should be pointed out. 

→ Before a frog becomes a tadpole, each developmental stage can be subdivided into Gosner stages. After metamorphosis, frog can be divided into froglets, juveniles, sub-adults, and adult frogs. Most studies of changes in skull shape have been conducted during stage of metamorphosis. It may be affected sufficiently at other times, but it has not been studied, so we did not explain it.

  1. English should be improved

→ We received English proofreading by native speakers. We can provide proof documents if this is required.

Round 2

Reviewer 2 Report

The authors have made changes that seem beffitting the comments. I will still say though that science works on the process of hypothesis testing and this study does not provide a hypotheses. Even if the hypotheses was that there was an expectation that the subdivision of the water catchment may correspond with a variation in genetic structure....for whatever reason. And just state that the catchment has a range of potential isolating structures but the understanding of the constraints are limited for this sepcies. Otherwise this is a 'look and see' project which in my mind is not justifed well enough especially when culling animals. in saying all that the technical aspects of this study are done well. I would suggest in the future though that the authors spend more time on developing the quesitons and hypotheses to create more compelling arguments for why the study is necessary. In some of the addtioinal text there are some typos and grammar issues - only minor but worth getting someone to glance over it again.